# Pathological Evidence for Residual SARS-CoV-2 in the Micrometastatic Niche of a Patient with Ovarian Cancer

**Takuma Hayashi** [1,2,3,*] 🔾, **Kenji Sano** [3,4], **Nobuo Yaegashi** [2,3,5] **and Ikuo Konishi** [1,2,3,6]

1    National Hospital Organization Kyoto Medical Centre, Kyoto 612-8555, Japan
2    The Japan Agency for Medical Research and Development (AMED), Tokyo 100-0004, Japan
3    PRUM-iBio Study Group, National Hospital Organization Tokyo Headquarter, Tokyo 100-0004, Japan
4    Department of Pathology, Shinshu University Hospital, Nagano 390-8621, Japan
5    Department of Obstetrics and Gynecology, Tohoku University Graduate School of Medicine, Miyagi 980-8575, Japan
6    Kyoto University Graduate School of Medicine, Kyoto 606-8501, Japan
\*    Correspondence: yoyoyo224@hotmail.com

**Abstract:** In previous clinical studies, severe acute respiratory syndrome coronavirus 2 (SARS-CoV-2) infection in cancer patients has a high risk of aggravation and mortality than in healthy infected individuals. Inoculation with coronavirus disease 2019 (COVID-19) vaccine reduces the risk of SARS-CoV-2 infection and COVID-19 severity. However, vaccination-induced anti-SARS-CoV-2 antibody production is said to be lower in cancer patients than in healthy individuals. In addition, the rationale for why the condition of patients with cancer worsens with COVID-19 is not well understood. Therefore, we examined the infection status of SARS-CoV-2 in the primary tumor and micrometastasis tissues of the patient with cancer and COVID-19. In this study, the expression of angiotensin-converting enzyme 2 (ACE2) was observed, and SARS-CoV-2 particles was detected in ovarian tissue cells in contact with the micrometastatic niche of the patient with high-grade serous ovarian cancer. We believe that the severity of COVID-19 in patients with cancer can be attributed to these pathological features. Therefore, the pathological findings of patients with advanced and recurrent ovarian cancer infected with SARS-CoV-2 may help decrease COVID-19 severity in patients with other cancer types.

**Keywords:** SARS-CoV-2; COVID-19; micrometastatic niche; patient with cancer



## 1. Introduction

The World Health Organization reports that the mortality rate of patients with cancer infected with SARS-CoV-2 is 7.6%, which is a fairly high rate compared with the 1.4% mortality rate of SARS-CoV-2-infected individuals without complications [1]. The 30 day all-cause mortality was high and associated with general and cancer-specific risk factors among patients with cancer and coronavirus disease 2019 (COVID-19), with a mortality of 13.3% [1–3]. COVID-19 severity in patients with cancer is yet to be elucidated. In patients with cancer undergoing anticancer therapy, a reduced immunity may exist [4].

Patients with hematological malignancies or solid tumors who have been infected with SARS-CoV-2 are at an increased risk of thromboembolism and associated complications, such as pulmonary vascular occlusive thromboinflammatory syndrome [5]. For hospitalized patients with cancer, prophylaxis using low-molecular-weight heparin (LMWH) is recommended for cancer patients infected with SARS-CoV-2 [5]. COVID-19 vaccination in healthy individuals reduces COVID-19 severity. Host immune responses to the SARS-CoV-2 vaccine may be suboptimal in patients with solid tumors, whereas SARS-CoV-2 vaccination-induced antibody productions are further reduced in patients with hematological malignancies [6]. The seroconversion rate of patients with hematological malignancies after the second dose of

COVID-19 vaccine is approximately 80%, which is lower than the 98% seroconversion rate observed in solid tumor patients [6].

As a preventive measure against SARS-CoV-2 infection, hygiene measures, physical distancing, and wearing of face masks are recommended for patients with cancer, intimate family members, and caregivers [5]. Furthermore, there may be a need to apply common preventive strategies, such as letting cancer patients stay in private rooms (a). To reduce clinic visits, routine SARS-CoV-2 swab testing, vaccination of healthcare workers working closely with cancer patients as well as caregivers, and efforts to reorganize the hematology unit with telemedicine are supported [5]. Summarizing the results of clinical studies so far, patients with cancer, especially those with hematological malignances, have a higher risk of developing COVID-19 and more severe outcomes. Hygiene precautions should be taken as early as possible during pandemic [7].

The pathological features of COVID-19, especially in patients with recurrent cancer or patients with cancer with relapse or metastasis, remain largely unknown. The cancer cells reportedly drive neighbouring differentiated alveolar type 2 cells to take on a stem-cell-like fate in micro metastatic niche [8–11] (Supplementary Table S1). Furthermore, lung organoid scRNA-seq at time points before ciliated differentiation identified expression of the SARS-CoV-2 receptor ACE2 and processing protease TMPRSS2 mRNAs predominantly in progenitor or stem-like cells of alveolar type 2 [12] (Supplementary Table S1). In our recent study, expression of angiotensin-converting enzyme 2 (ACE2), a host-side receptor for SARS-CoV-2, was observed in ovarian tissue cells in contact with the micrometastatic niche of high-grade serous ovarian cancer (HG-SOC) [13]. In addition, the possibility of SARS-CoV-2 infection was observed in ovarian tissue cells in which such an ACE2 expression was detected [13]. Our research aimed to identify the cause of increasing COVID-19 severity in cancer patients using molecular pathological studies. The primary endpoint of our clinical research is to show that the severity of COVID-19 in cancer patients can be attributed to these pathological features.

## 2. Materials and Methods

### 2.1. Antibodies

The list of antibodies, which were used as the first monoclonal antibody or secondary antibody in our immunohistochemistry (IHC) research experiments, is shown in the Materials and Methods section in the Supplementary Information.

### 2.2. Immunohistochemistry

IHC staining for CD90, S100A4, ACE2, and receptor binding domain (RBD) of SARS-CoV-2 spike glycoprotein was performed on tissue sections of HG-SOC. Antibodies for CD90 (Thy1) (ab133350) and S100A4 (ab124805) were purchased from Abcam Inc. (Cambridge, UK). RBD of the spike glycoprotein was purchased from GeneTex Inc. (Irvine, CA, USA). 4′,6-Diamidino-2-phenylindole (DAPI) mounting medium was purchased from VECTOR LABORATORIES, Inc. (Burlingame, CA, USA). IHC was performed using normal methods with the primary antibody and second antibody conjugated with immunofluorescence as described previously [13]. Details of the IHC experiment are indicated in the Materials and Methods section in the Supplementary Information.

### 2.3. Transmission Electron Microscopy

TEM was performed under a routine procedure. Tissues were fixed in 10% formaldehyde for 1 day. Briefly, specimens (approximately 1 mm × 1 mm × 1 mm in size) from each organ were fixed in 2.5% glutaraldehyde in 0.1 M phosphoric buffer (pH: 7.4) for 24 h, postfixed with 1% osmium tetroxide, dehydrated with gradient alcohol, and embedded using Eponate 12™ Kit with DMP-30 (18010, TED PELLA Inc., Redding, CA, USA). Details of the IHC experiment are indicated in the Materials and Methods section in the Supplementary Information.

*2.4. Statistical Analysis*

All data are expressed as the mean and standard error of the mean. Normality was verified using the Shapiro–Wilk test. For comparing two groups, the unpaired two-tailed *t* test or Mann–Whitney *U* test was used. Multiple comparisons were performed using a one-way analysis of variance with a Tukey post hoc test or a Kruskal–Wallis analysis with a post hoc Steel–Dwass or Steel test as described previously [14]. A *p*-value of <0.05 was considered statistically significant. All statistical analyses were conducted using the JMP software (SAS Institute, Cary, NC, USA).

*2.5. Institutional Review Board Statement and Consent to Participate*

Details of Materials and Methods are indicated in Supplementary Materials.

**3. Case Report**

A 47-year-old woman was admitted to the hospital on 18 May 2022, because of trauma related to a fall. This patient reported that she had been exposed to a COVID-19 patient on 15 May 2022. Since the exposure, the patient manifested pneumonia symptoms (Supplementary Table S2). On 15 May 2022, the patient underwent a nasopharyngeal swab and was confirmed positive for SARS-CoV-2 via a reverse transcription-polymerase chain reaction (RT-PCR) test followed by treatment [15]. She was sent home for treatment with an antipyretic and cough medicine.

At approximately 11:00 a.m. on 18 May 2022, the patient complained of severe lower abdominal pain. The patient was examined at a nearby general hospital. Computed tomography (CT) was performed and twisting of a left ovarian tumor with a maximum diameter of 13 cm was suspected. Because of the urgent need for surgical treatment, the patient was referred to our institution for an emergency transfer (Supplementary Table S1). After completing the transfer procedures for COVID-19 patients, the patient was taken to our hospital, which is the designated national hospital for accepting highly acute phase/tertiary critical care/COVID-19 severely ill patients, via ambulance (Supplementary Table S1).

On May 18, a chest X-ray was performed, which showed scattered ground glass-like shadows in the lower left lung field and a granular shadow in the right lung field (Supplementary Figure S1A). Sosegon Intravenous Solution 15 mg, an analgesic, was administered for the lower abdominal pain and sore throat on 18 May at 10:50 p.m.; however, the pain persisted; hence, Acerio Intravenous Solution 1000 mg was administered (Supplementary Table S1).

On 18 May, a CT scan revealed a cystic mass with a major axis of approximately 12 cm in the left ovary (Supplementary Figure S1B). The ovarian stroma was edematous. Images taken by the coronal section showed a spiral structure between the uterus and the lesion in the left ovary. The possibility of left ovarian tumor volvulus was primarily considered. A malignant ovarian tumor was suspected because a solid mass was found in the left ovarian tissue (Supplementary Figure S1B). The right ovary revealed unremarkable CT findings. Furthermore, fatty liver was also noted, but with no significant lymphadenopathy or ascites. Additionally, the CT findings suggested left malignant ovarian tumor pedicle torsion.

On May 19 at 01:52 a.m., laparoscopic left and right uterine adnexal tumor resection was performed (Laparoscopic uterine adnexal tumor resection: The patient was placed in the supine position, and a 12 mm port was inserted in the navel using an open method, followed by the placement of a 5 mm port in the lower left, midline, and lower right areas of the abdomen. Surgery was performed using the diamond method. Using Enseal, we dissected the left and right fallopian tubes, left and right pelvic funnel ligament, and mesovarium and removed the left and right adnexa (ovary and fallopian tube). After aspirating the contents of the surgical area, we placed the excised specimen in a collection bag for examination. We confirmed hemostasis, cleaned the abdominal cavity, and removed the port. The peritoneum was sutured using Vicryl 2-0, and the dermis was sutured with Mono-Dox 4-0) secondary to malignant left ovarian tumor torsion. The right ovary and fallopian tubes were intact. The left appendage (ovary and fallopian tube) was twisted 540° clockwise. The size of the suspected

malignant ovarian tumor resembled a newborn head. Laparoscopic examination showed minimal ascites, but with no adhesions. On 20 May, a decrease in blood oxygen concentration (91%) was observed while the patient was asleep; thus, oxygen inhalation was initiated with 1 L of oxygen per minut (Supplementary Table S1).

Pathological findings: To evaluate the resected ovaries and fallopian tubes, the fallopian tubes were longitudinally incised as per protocol and the fimbriated end (SEE-Fim) was extensively examined. The remaining ovaries and fallopian tubes were cut into 2–3 mm sections. Histopathological evaluation was conducted (Supplementary Figure S2). High-grade serous carcinoma was found in the papillary nodule on the right outer surface of the left ovary. Serous tubal intraepithelial carcinoma (STIC) and high-grade serous carcinoma are found in the left fimbriated end (Figure 1). No malignant findings were found in the right ovary and fallopian tube. From the immunohistochemical staining results, tumor protein 53 (TP53)-positive and Ki-67/MIB1-strongly positive highly cellular atypical and nuclear atypical cells are observed in the left fimbriated end. In the papillary nodule tissue found on the right outer surface of the left ovary, paired box gene 8 (PAX8)-positive, Wilms tumor-1 (WT-1)-positive, and TP53-diffusely strongly positive highly cellular atypical and nuclear atypical cells are observed (Figure 1, Supplementary Figure S2B).

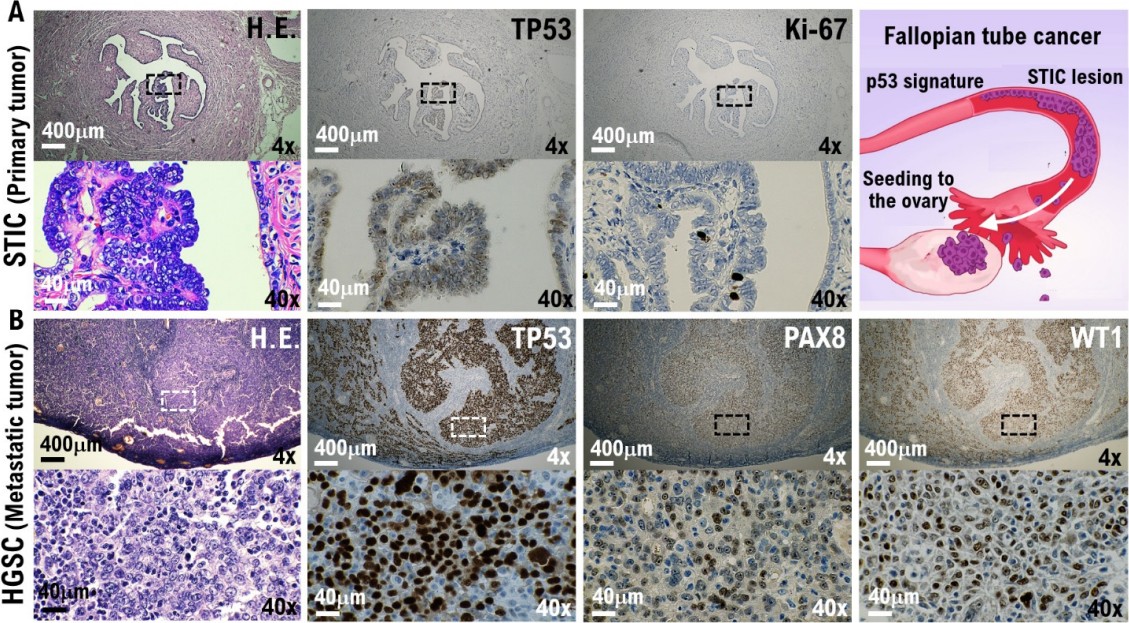

**Figure 1.** Molecular pathological evidence for the development of high-grade serous ovarian cancer (HG-SOC) in a patient with COVID-19. Molecular pathological analysis was conducted on the bilateral fallopian tubes and ovarian tissues removed from COVID-19 patients. To evaluate the resected ovaries and fallopian tubes, the fallopian tubes were longitudinally incised as per protocol and the fimbriated end (SEE-Fim) was extensively examined. The remaining ovaries and fallopian tubes were cut into 2–3 mm sections. Histopathological evaluation was conducted using these tissue sections. (**A**) The H.E. images show papillary and dendritic lesions forming in part of the fallopian tube. Enlargement of the image showing the diseased tissue reveals noninvasive growth of strongly atypical epithelium (H.E.). In the immunohistochemical staining results with appropriate monoclonal antibodies, TP53-positive and Ki-67/MIB1-strongly positive highly cellular atypical and nuclear atypical cells are observed in the left fimbriated end. Serous tubal intraepithelial carcinoma (STIC) and HG-SOC are found in the left fimbriated end. (**B**) The H.E. image shows growing solid tissue areas and papillary structures. Enlarging the lesion tissue reveals a narrow gap within the solid alveolus (H.E.). In the papillary nodule found on the right outer surface of the left ovary, PAX8-positive, WT-1-positive, and TP53-diffusely strongly positive highly cellular atypical and nuclear atypical cells are observed. High-grade serous carcinoma was found in the papillary nodule on the right outer surface of the left ovary. As shown in the panels, scales are 4× and 40×.

Previous studies have revealed that the origin of HG-SOC is STIC and/or epithelial malignancies in the fallopian tubes. STIC could be a likely precursor lesion of high-grade serous pelvic carcinomas, carcinosarcoma, and undifferentiated carcinoma with an incidence of 0.6–7% in breast cancer 1/2 gene carriers or women with a strong family history of breast or ovarian carcinoma. Therefore, epithelial malignant cells and STICs that occur in the epithelial cell tissue of the fallopian tubes and in the fimbriated end are the primary tumors that metastasize into the ovary and form micrometastases [16–18]. In forming a micrometastatic niche in multiple organs, epithelial and stromal cells in contact with circulation tumor cells (CTCs) are initialized by secretory factors from CTCs that compose the micrometastatic niche [8–11]. Furthermore, lung organoid scRNA-seq at time points before ciliated differentiation identified expression of the SARS-CoV-2 receptor ACE2 and processing protease TMPRSS2 mRNAs predominantly in progenitor or stem-like cells of alveolar type 2 [12] ACE2, a host-side receptor for SARS-CoV-2, was expressed in CD90-positive alveolar epithelial stem-like cells in the pulmonary metastatic niches of no-SARS-CoV-2-infected 5 patients with high-grade serous ovarian cancer is essential (Table 1). Furthermore, histopathological analyses showed that the RBD of the SARS-CoV-2 spike glycoprotein bound to ACE2-expressing CD90-positive alveolar epithelial stem-like cells (Table 1). Based on these findings, SARS-CoV-2 is deemed to infect the alveolar epithelial stem-like cells in pulmonary micrometastases of patients with ovarian cancer. ACE2 was possibly expressed in epithelial stem-like cells or progenitor cells [12,13]. Additionally, SARS-CoV-2 infection of the epithelial cells and stromal cells in contact with or in the micrometastatic niche composed of HG-SOC found in the left ovary was considered. We investigated SARS-CoV-2 infection in the micrometastatic niche composed of HG-SOC using a molecular pathological method from the left ovarian tumor tissue resected from a COVID-19 patient.

In the IHC results using the anti-S100A4 antibody, which is a biomarker of HG-SOC, the cell population with a high degree of cell atypia and nuclear atypia observed in the left ovary was confirmed to be HG-SOC cells (Figure 2A). In the IHC results using anticyclin-dependent kinase 15 (CDK15) antibody, which is a connective tissue biomarker (stroma cells), ovarian connective tissue (stroma cells) was confirmed (Figure 2A, Supplementary Figure S10). A strong expression of ACE2 and cluster of differentiation 90 (CD90), which are biomarkers of stem-like cell/progenitor cell, was also observed in the ovary in some cells of the connective tissue/stroma cells in contact with or in the vicinity of the micrometastatic niche of HG-SOC (Figure 2B, Supplementary Figure S3). Conversely, ACE2 expression was not observed in the connective tissue cells (stroma cells) other than the micrometastatic niche (Figure 2B, Supplementary Figure S3). Furthermore, we conducted IHC staining using a monoclonal antibody against SARS-CoV-2 spike glycoprotein, and it was found that SARS-CoV-2 may be present in the HG-SOC tissue (Figure 2B). From IHC results using the antispike glycoprotein of SARS-CoV-2 antibody, spike protein (Spike P.) was observed in and around the cells of the connective tissue/stroma cells positive for ACE2 expression (Figure 2B, Supplementary Figure S3). Furthermore, we conducted a molecular histopathological examination using transmission electron microscopy (TEM). Consistently, images derived from observation using TEM showed clear SARS-CoV-2 particles in some cells of the connective tissue/stroma cells in contact with or in the vicinity of the micrometastatic niche of HG-SOC found in the ovary (Figure 3). As already reported [19], a heterogeneous, electron-dense, partly granular interior with ribonucleoprotein can be differentiated (Figure 3F, black arrowhead), envelope membranes of coronavirus are well resolved, and some particles show delicate surface projections (i.e., spikes; Figure 3F; white arrowhead). Thus, SARS-CoV-2 infection was confirmed in some cells of the connective tissue/stroma cells in contact with or in the vicinity of the micrometastatic niche of HG-SOC found in the ovary, and SARS-CoV-2 budding was confirmed (Figure 3). The viral particles measured 80–100 nm in diameter (Figure 3). By contrast, the expression of ACE2 and spike protein was not observed in the cortex region in the excised tissue, follicle cells, and stroma cells in tissue array sections (Supplementary Figures S4 and S5).

**Table 1.** Characteristics of patients with ovarian cancer and lung metastases as well as CD90 and ACE2 expression in the metastasis areas and the alveolar and bronchiolar areas.

| Patient No. | Age Range | Age at Surgery (Years) | Histological Type | FIGO Stage | Grade | No. of Lung Metastatic Lesions | Pulmonary Metastatic Niche | | Vital Status |
|---|---|---|---|---|---|---|---|---|---|
| | | | | | | | CD90* (%) | ACE2* (%) | |
| 1 | 40s | 40–45 | HG serous | IVA | 3 | Single | 36.43 | 27.42 | Alive |
| 2 | 50s | 50–55 | HG serous | IVA | 3 | Single | 33.87 | 18.93 | Alive |
| 3 | 50s | 50–55 | HG serous | IVA | 3 | Multiple | 38.32 | 29.38 | Deceased |
| 4 | 40s | 45–50 | HG serous | IVB | 3 | Multiple | 32.67 | 28.05 | Alive |

| Patient no. | Age range | Age at surgery (years) | Histological type | FIGO stage | Grade | No. of metastatic lesions | Ovarian metastatic niche | | Vital status |
|---|---|---|---|---|---|---|---|---|---|
| | | | | | | | CD90* (%) | ACE2* (%) | |
| 5 | 40s | 45–50 | HG serous | IIA | 2 | Single | 16.33 | 15.34 | Alive |

**Normal alveolar and bronchiolar areas.**

| Patient no. | Normal alveoli | | Normal bronchioles | | SARS-CoV-2[a] | COVID-19[b] |
|---|---|---|---|---|---|---|
| | CD90* (%) | ACE2* (%) | CD90* (%) | ACE2* (%) | | |
| **1** | **4.53** | 11.82 | 3.23 | 20.67 | Negative | Negative |
| 2 | 3.91 | 12.57 | 3.18 | 21.46 | Negative | Negative |
| 3 | 4.34 | 12.71 | 3.45 | 22.05 | Negative | Negative |
| 4 | 4.08 | 13.43 | 2.98 | 21.92 | Negative | Negative |
| 5[c] | NA | NA | NA | NA | Positive | Positive |

FIGO stage, the FIGO (International Federation of Gynecology and Obstetrics) staging system is commonly used for cancers of the female reproductive organs. High grade (HG) serous, high-grade serous ovarian adenocarcinoma. CD90*, proportion of CD90-positive alveolar epithelial stem-like cells in pulmonary metastatic niches, normal alveoli, and bronchioles assessed by immunohistochemical experiments using antihuman CD90 monoclonal antibody. ACE2*, proportion of ACE2-positive alveolar epithelial stem-like cells in pulmonary metastatic niches, normal alveoli, and bronchioles assessed by immunohistochemical experiments using antihuman ACE2 monoclonal antibodies. The expression levels of each factor were determined by measuring the fluorescence intensities. Percentages are the ratio of CD90 or ACE2-positive cells to the total cell counts. NA; no answer. SARS-CoV-2[a]; SARS-CoV-2 positive by PCR-examination, COVID-19[b]; Diagnosed with COVID-19 by X-ray and CT imaging. Patient #5[c]; Patient #5 is the COVID-19-positive individual reported in this research manuscript.

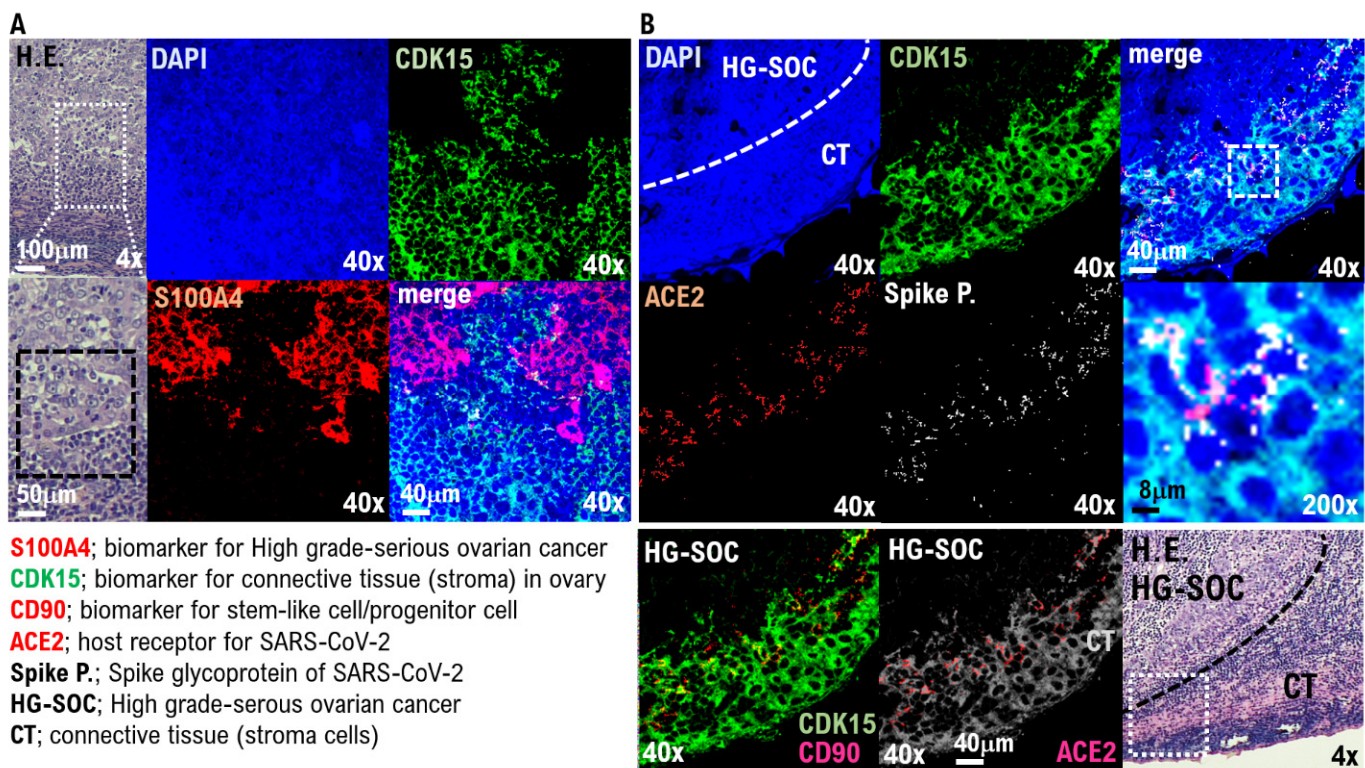

**Figure 2.** SARS-CoV-2 infection in the connecting tissue in contact with or in the vicinity of HG-SOC micrometastatic niche found in the ovary. SARS-CoV-2 infection in the micrometastatic niche composed of HG-SOC was examined through a molecular pathological method using the left ovarian tumor tissue resected from COVID-19 patients. (**A**) The H.E. image shows growing solid tissue areas and papillary structures. Enlarging the lesion tissue reveals a narrow gap within the solid alveolus (H.E.). From the immunohistochemistry (IHC) staining results using the anti-S100A4 antibody, which is a biomarker of HG-SOC, the cell population with a high degree of cell atypia and nuclear atypia observed in the left ovary was confirmed to be HG-SOC cells. From the IHC staining results using an anticyclin-dependent kinase 15 (CDK15) antibody, which is a connective tissue biomarker (stroma cells), the connective tissue of the ovary was markedly detected. (**B**) The H.E. image shows the boundary between the HG-SOC tissue and the connecting tissue (CT) of the ovary. A strong expression of ACE2 and CD90, which are biomarkers for stem-like cells/progenitor cells, was observed in some cells of the connecting tissue in contact with or in the vicinity of the micrometastatic niche of HG-SOC found in the left ovarian tissue. Immunohistochemistry staining was performed using a monoclonal antibody against SARS-CoV-2 spike glycoprotein. The spike glycoprotein was clearly detected, suggesting that the SARS-CoV-2 virus may be present in the HG-SOC tissue. From the IHC staining results using the antispike glycoprotein of SARS-CoV-2 monoclonal antibody, spike protein (Spike P.) was observed in and around the cells of the connecting tissue positive for ACE2 expression.

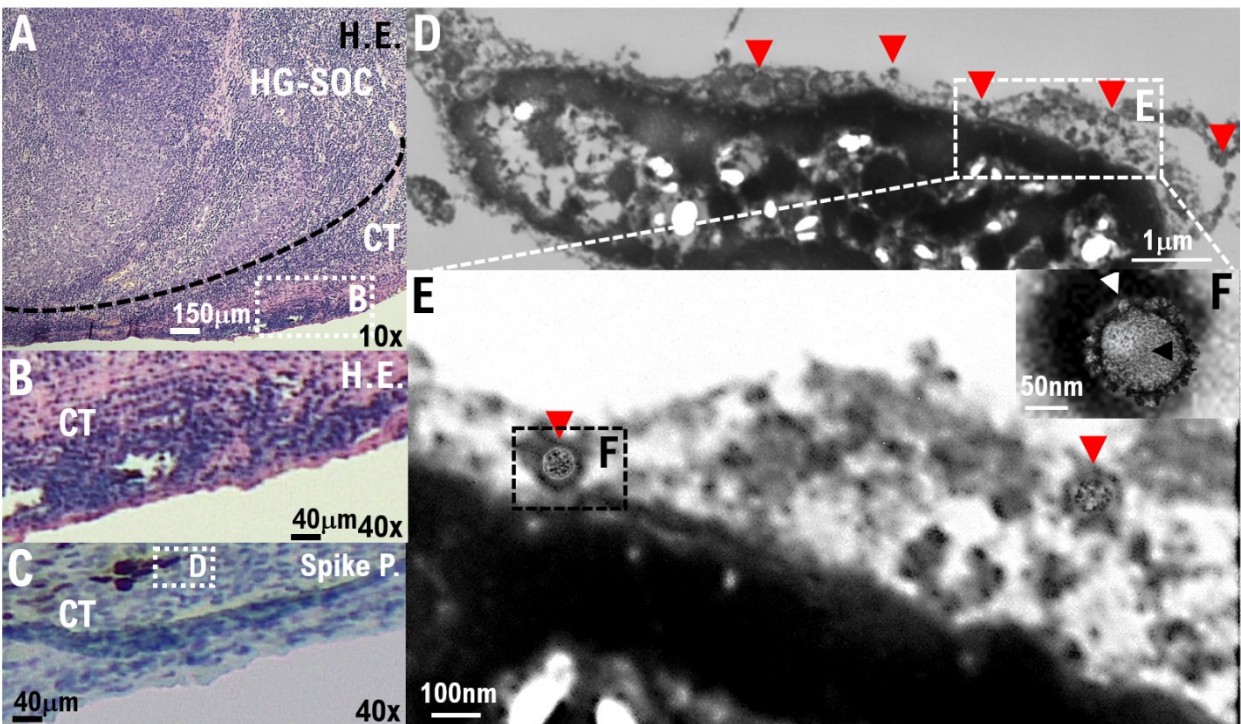

**Figure 3.** SARS-CoV-2 infection of the connecting tissue in contact with or in the micrometastatic niche of HG-SOC using transmission electron microscopy (TEM). (**A,B**) Hematoxylin and Eosin (H.E.) staining photographs show some cells of the connecting tissue (stroma cells) in contact with or in the micrometastatic niche of HG-SOC found in the ovary. (**C**) We conducted IHC staining using a monoclonal antibody against SARS-CoV-2 spike glycoprotein and confirmed that the spike glycoprotein of SARS-CoV-2 existed in some cells of the connecting tissue (stroma cells) in contact with or in the micrometastatic niche of HG-SOC found in the ovary. (**D–F**) We conducted a molecular histopathological examination with TEM. Consistently, images using electron microscopic observation showed clear SARS-CoV-2 particles in some cells of the connecting tissue (stroma cells) in contact with or in the micrometastatic niche of HG-SOC found in the ovary. The findings obtained from histopathological examination using TEM indicate a heterogeneous, electron-dense, partly granular interior with a ribonucleoprotein that can be differentiated (**F**, black arrowhead), coronavirus envelope membranes are well resolved, and some particles show delicate surface projections (i.e., spikes; **F**, white arrowhead). Scales are 10,000× (**D**), 20,000× (**E**), and 50,000× (**F**).

## 4. Discussion

The pathological features of COVID-19, especially in recurrent cancer tissues or metastatic tissues in patients with relapse or metastasis, remain largely unknown. In this study, a molecular pathological examination was conducted on HG-SOC excised tissue of an ovarian metastasis from a primary fallopian tube cancer from a patient with mild COVID-19 pneumonia. In the excised cancer tissue, changes in the molecular pathology were discovered, which were attributed to the SARS-CoV-2 infection. SARS-CoV-2 proliferating in the cancer tissue of patients with HG-SOC was identified through a comprehensive examination using electron microscopy and IHC staining (Figure 3, Table 1). It should be noted that ACE2 expression was observed, and SARS-CoV-2 particles were detected in ovarian histiocytes in contact with the micrometastatic niche of HG-SOC. Factors secreted from the micrometastatic niche of HG-SOC may induce SARS-CoV-2 infection and proliferation.

A recent report demonstrated that neither coronavirus particles nor SARS-CoV-2 nucleocapsid was detected in the liver, heart, intestine, skin, and bone marrow [20]. A previous study highlighted that SARS-CoV-2 remnants in the lung of a discharged COVID-19 patient after a series of nasopharyngeal swabs confirmed via RT-PCR showed negative results for SARS-CoV-2 [20]. Moreover, our research results revealed that SARS-CoV-2 is proliferating

in cancer tissues in patients with cancer symptomatic of COVID-19. Therefore, compared with patients infected with COVID-19 with no comorbidities, patients with cancer and COVID-19 may have more SARS-CoV-2 proliferating in the liver, heart, intestines, skin, and bone marrow. Based on our research results, the excision of metastatic tissues in distant organs in patients with cancer and COVID-19 is considered to be an important intervention to prevent the aggravation of COVID-19.

Recent research demonstrated that ovarian cancer is associated with immune deficiencies leading to tumor progression in the host. These effects are associated with the presence of regulatory T cells, inhibition of natural killer cytotoxic responses, accumulation of myeloid suppressor cells in the tumor, deficiencies on interferon signaling, secretion of cytokines that enhance tumor growth (e.g., IL-6, IL-10, CSF-1, TGF-β, TNF), and expression of surface molecules (e.g., HLA-G, B7-H1, B7-H4, CD40, CD80) that play a role in immune suppression [21]. Presumably, the weakened immune function observed in ovarian cancer patients affects the growth of ovarian cancer [22,23] and, in turn, the infectivity of SARS-CoV-2 to ovarian cancer patients. On the other hand, our pathological studies demonstrated the increased expression of ACE2 and SARS-CoV-2 virus particles in the vicinity of ovarian tissue cells that are in contact with the micrometastatic niche. The cytokines secreted by ovarian tissue cells that are in contact with the micrometastatic niche are thought to induce upregulation of ACE2 expression. Further clinical studies are required to elucidate the association between cytokine expression observed in patients with ovarian cancer and SARS-CoV-2 infectivity.

The severity rate of patients with cancer and COVID-19 is clearly high compared with patients with COVID-19 with no underlying disorders [1]. This may be attributed to the low immunity among patients with cancer; however, the exact mechanism is yet to be clarified. Our study provided pathological evidence of SARS-CoV-2 proliferation in metastatic lesions in organs of patients with cancer and COVID-19. Although surgical resection was performed in the cancerous tissue, SARS-CoV-2 proliferates in the lungs, liver, heart, intestines, skin, and bone marrow. Since clinical studies to date have shown that the third dose of the COVID-19 vaccine reduces the morbidity and mortality of COVID-19 in patients with cancer, advocating for COVID-19 vaccination to patients with cancer is necessary [24,25]. Additionally, therapeutic agents such as antiviral antibody drugs should be administered to patients with cancer in the early stage of SARS-CoV-2 infection [26]. Our research demonstrated the increased expression of ACE2 and SARS-CoV-2 virus particles in the vicinity of ovarian tissue cells that are in contact with the micrometastatic niche. The pathological findings of patients with advanced and/or recurrent ovarian cancer infected with SARS-CoV-2 may help decrease COVID-19 severity in patients with other cancer types. In clinical studies, molecular pathological analysis in patients infected with COVID-19 with other cancer types is needed. Moreover, a timely follow-up of health examinations for patients with cancer is strongly recommended in clinical practice.

**Supplementary Materials:** The following supporting information can be downloaded at: https://www.mdpi.com/article/10.3390/cimb44120400/s1, Figure S1: Suspicion of left ovarian tumor volvulus in a patient with COVID-19. Figure S2: Detection of p53-positive cells in the left ovarian tissue derived from a patient with COVID-19. Figure S3: Detection of SARS-CoV-2 spike glycoprotein (Spike P.) in some cells of the connecting tissue in contact with or in the micrometastatic niche of HG-SOC. Figure S4: No ACE2 and spike glycoprotein of SARS-CoV-2 expression in the normal ovarian tissue. Figure S5: No ACE2 expression in normal ovarian tissues. Figure S6: No expression of ACE2 and SARS-CoV-2 spike glycoprotein in the normal tissue region of the fallopian tubes derived from the patient. Figure S7: No expression of ACE2 in the normal fallopian tube tissues. Figure S8: The mechanism of changes in the molecular pathology attributed to the SARS-CoV-2 infection. Figure S9: SARS-CoV-2 infection in ovarian histiocytes in contact with the micrometastatic niche of HG-SOC. Figure S10: The expression of CDK15 in connective tissue/stroma cells; no expression of CDK15 in serous ovarian cancer. Table S1: Reports published so far. Table S2: Clinical description and test for the patient.

**Author Contributions:** T.H. and K.S. performed most of the clinical work and coordinated the project. T.H. and K.S. conducted the diagnostic pathological studies. T.H. and K.S. conceptualized the study and wrote the manuscript. T.H., K.S. and N.Y. and carefully reviewed this manuscript and commented on the aspects of medical science. I.K. shared information on clinical medicine and oversaw the entirety of the study. All authors have read and agreed to the published version of the manuscript.

**Funding:** Japan Society for Promoting Science: 19K09840; START-program Japan Science and Technology Agency (JST): STSC20001; National Hospital Organization Multicenter clinical study: 2019-Cancer ingeneral-02.

**Institutional Review Board Statement:** These experiments with human tumor tissues derived from patients with high grade-serous ovarian cancer were conducted at Shinshu University and National Hospital Organization Kyoto Medical Center in accordance with institutional guidelines (i.e., IRB approval no. M192, H31-cancer-2). IRB no M192 was approved at date: 5 April 2014, and 16 June 2016. IRB no H31-cancer-2 was approved at date: 9 November 2019, and 17 June 2022. The authors attended research ethics education through the Education for Research Ethics and Integrity (APRIN e-learning program (eAPRIN)). The completion numbers for the authors are AP0000151756, AP0000151757, AP0000151769, and AP000351128. Consent to participate was required as this research was considered clinical research. Subjects signed an informed consent form when they were briefed on the clinical study and agreed with content of the research.

**Informed Consent Statement:** Informed consent was obtained from all subjects involved in the study.

**Data Availability Statement:** All data are shown in the manuscript figures and Supplementary Information. All the data generated in this study are provided in the Source Data file.

**Acknowledgments:** We appreciate David Baltimore (Nobel Laureate, California Institute of Technology) for biological comments. We thank all medical staff for providing animal care at Shinshu University School of Medicine and the National Hospital Organization Kyoto Medical Center. This clinical research was performed with research funding from the following: Japan Society for Promoting Science for TH (Grant No. 19K09840), for KA (No. 20K16431), and START-program Japan Science and Technology Agency for TH (Grant No. STSC20001), and the National Hospital Organization Multicenter clinical study for TH (Grant No. 2019-Cancer in general-02).

**Conflicts of Interest:** The authors declare no potential conflict of interest.

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
