# Peer review of "Pathological Evidence for Residual SARS-CoV-2 in the Micrometastatic Niche of a Patient with Ovarian Cancer"

_cimb, doi:10.3390/cimb44120400_

Round 1
Reviewer 1 Report (Previous Reviewer 3)
The work has been improved and it is suitable for publication
Author Response
Manuscript ID: cimb-2058323
Reviewer 1
Comment. The work has been improved and it is suitable for publication
Answer. We appreciate your positive comment for publishing our manuscript.

Reviewer 2 Report (New Reviewer)
1. Figure caption 1 related to H.E. is not mentioned. A and B need to be bold. Revise other related figures.
2. Some texts in several parts have different sizes or spaces. Please re-check.
3. Add clearer scales with size in all figures, including supplementary figures (not necessary to use a ruler as scale).
4. Red font number in S.Figure 2 is unclear.
5. Several figures in supplementary have unclear text. Please use clearer text or resize the figures.
6. I think adding an illustrative figure to explain the mechanism of changes in the molecular pathology attributed to the SARS-CoV-2 infection is necessary. It can be placed in the discussion section.
7. Adding a table that resumes similar (or quite related) cases in the discussion section is also necessary to help the readers know the issues on other reports. Information on previous reports also needs to be added in the introduction and discussion sections. The discussion section could be improved deeper based on this.
Author Response
Manuscript ID: cimb-2058323
Reviewer 2
Comment 1. Figure caption 1 related to H.E. is not mentioned. A and B need to be bold. Revise other related figures.
Answer 1. We appreciate and agree with the comments from the reviewer. We have revised our manuscript according to the reviewer’s instructions as follows.
I made the A and B bold.
Figure 1. Molecular pathological evidence for the development of high-grade serous ovarian cancer (HG-SOC) in a patient with COVID-19. Molecular pathological analysis was conducted on the bilateral fallopian tubes and ovarian tissues removed from COVID-19 patients. To evaluate the resected ovaries and fallopian tubes, the fallopian tubes were longitudinally incised as per protocol and the fimbriated end (SEE-Fim) was extensively examined. The remaining ovaries and fallopian tubes were cut into 2–3 mm sections. Histopathological evaluation was conducted using these tissue sections. A. The H.E. images show papillary and dendritic lesions forming in part of the fallopian tube. Enlargement of the image showing the diseased tissue reveals noninvasive growth of strongly atypical epithelium (H.E.). In the immunohistochemical staining results with appropriate monoclonal antibodies, TP53-positive and Ki-67/MIB1-strongly positive highly cellular atypical and nuclear atypical cells are observed in the left fimbriated end. Serous tubal intraepithelial carcinoma (STIC) and HG-SOC are found in the left fimbriated end. B. The H.E. image shows growing solid tissue areas and papillary structures. Enlarging the lesion tissue reveals a narrow gap within the solid alveolus (H.E.). In the papillary nodule found on the right outer surface of the left ovary, PAX8-positive, WT-1-positive, and TP53-diffusely strongly positive highly cellular atypical and nuclear atypical cells are observed. High-grade serous carcinoma was found in the papillary nodule on the right outer surface of the left ovary. As shown in the panels, scales are 4× and 40×.
Comment 2. Some texts in several parts have different sizes or spaces. Please re-check.
Answer 2. We appreciate and agree with the comments from the reviewer. We checked the entire manuscript for the points made by the reviewers. Thank you for your good advice again.
Comment 3. Add clearer scales with size in all figures, including supplementary figures (not necessary to use a ruler as scale).
Answer 3. We appreciate and agree with the comments from the reviewer. We added clearer scales with size in all figures, including supplementary figures.
Comment 4. Red font number in S.Figure 2 is unclear.
Answer 4. We appreciate the comments from the reviewer. We well understand the reviewer’s comment. We revised S.Figure 2 according to the reviewer’s instructions as follows. A pathologist wrote the name and number of the tissue site in red marker on each block of tissue removed by surgical treatment. Red lines and numbers indicate cross-sections in the gross findings of excised tissues. The numbers in red in the left panel of A match the numbers in red in the right panel of A.
- Figure 2. Detection of p53-positive cells in the left ovarian tissue derived from a patient with COVID-19. A. Cut surface of the excised fallopian tube and ovary: The mass found in the left ovary from a COVID-19 patient is a typical HG-SOC on macroscopic and microscopic examination. In the left ovary, a small grayish-white mass containing necrosis and hemorrhagic tissue is observed. Red lines and numbers indicate cross-sections in the gross findings of excised tissues. The numbers in red in the left panel of A match the numbers in red in the right panel of A. B. The left panel shows an H.E.-stained tissue section obtained from the left ovary of a patient with COVID-19. Recent research demonstrated that p53 is one of the biomarkers for HG-SOC. Therefore, a molecular pathological study was conducted, and the IHC study with antihuman p53 monoclonal antibody shows that p53-positive cells are observed in tumor-like mass in the left ovary (center panel and right panel). The results of this study confirmed the presence of an HG-SOC in the left ovary. Scale: 10×.
Comment 5. Several figures in supplementary have unclear text. Please use clearer text or resize the figures.
Answer 5. We appreciate and agree with the comments from the reviewer. We have created and revised Supplement Figure 5, Supplement Figure 7, and Appendix Figure 1 according to the reviewer’s instructions.
Comment 6. I think adding an illustrative figure to explain the mechanism of changes in the molecular pathology attributed to the SARS-CoV-2 infection is necessary. It can be placed in the discussion section.
Answer 6. We appreciate and agree with the comments from the reviewer. We added an illustrative figure to explain the mechanism of changes in the molecular pathology attributed to the SARS-CoV-2 infection is necessary. We created an additional illustrative figure as New Supplement Figure 8 as following.
- Figure 8. The mechanism of changes in the molecular pathology attributed to the SARS-CoV-2 infection. In the micrometastatic niche, epithelial cells etc. that are in contact with cancer cells are differentiated into stem-like cells and progenitor cells by some cytokine(s) or other factor(s). ACE2, the receptor for SARS-CoV-2 markedly expresses in CD90-positive stem-like cells or CD90-positive progenitor cells that are in contact with metastatic tumors in organs and tissues that did not express ACE2. Therefore, for COVID-19 patients with metastasis to other organs, SARS-CoV-2 infection is observed in cells that are in contact with metastatic tumors in the micrometastatic niche.
Comment 7. Adding a table that resumes similar (or quite related) cases in the discussion section is also necessary to help the readers know the issues on other reports. Information on previous reports also needs to be added in the introduction and discussion sections. The discussion section could be improved deeper based on this.
Answer 7. We appreciate and agree with the comments from the reviewer. We created Table as Supplement Table 1 and added a New Supplement Table 1 that resumes similar (or quite related) cases in the discussion section is also necessary to help the readers know the issues on other reports. Information on previous reports also was added in the introduction and discussion sections in the revised manuscript.
S.Table 1. Reports published so far
|
Previously report |
Summary of report |
Ref. |
|
Shiozawa Y, Pedersen EA, Havens AM, Jung Y, Mishra A, Joseph J, Kim JK, Patel LR, Ying C, Ziegler AM, Pienta MJ, Song J, Wang J, Loberg RD, Krebsbach PH, Pienta KJ, Taichman RS. Human prostate cancer metastases target the hematopoietic stem cell niche to establish footholds in mouse bone marrow. J Clin Invest 2011; 121: 1298–1312. |
Competition between disseminated human prostate cancer cells and hematopoietic stem cells (HSC) for the endosteal niche facilitates metastasis. Metastatic cells shed from a primary tumor compete with HSCs to engage the endosteal niche, which suggests that solid tumors use the HSC niche as metastatic niche. they settle near stem cells in bone marrow, human prostate cancer cells promotes the development of a metastatic environment that supports tumor growth. |
8 |
|
Malanchi I, Santamaria-Martínez A, Susanto E, Peng H, Lehr HA, Delaloye JF, Huelsken J. Interactions between cancer stem cells and their niche govern metastatic colonization. Nature 2012; 481: 85-89. |
The cancer cells drive neighbouring differentiated alveolar type 2 cells to take on a stem-cell-like fate. Reports described cancer-associated parenchymal cells that exhibit stem cell-like features, the expression of lung progenitor markers, multi-lineage differentiation potential, and self-renewal activity. |
9 |
|
Ombrato L, Nolan E, Kurelac I, Mavousian A, Bridgeman VL, Heinze I, Chakravarty P, Horswell S, Gonzalez-Gualda E, Matacchione G, Weston A, Kirkpatrick J, Husain E, Speirs V, Collinson L, Ori A, Lee JH, Malanchi I Metastatic-niche labelling reveals parenchymal cells with stem features. Nature 2019; 572: 603-608. |
The cancer cells drive neighbouring differentiated alveolar type 2 cells to take on a stem-cell-like fate. Reports described cancer-associated parenchymal cells that exhibit stem cell-like features, the expression of lung progenitor markers, multi-lineage differentiation potential, and self-renewal activity. |
10 |
|
Hayashi T, Sano K, Aburatani H, Yaegashi N, Konishi I. Initialization of epithelial cells by tumor cells in a metastatic microenvironment. Oncogene 2020 Mar;39(12):2638-2640. |
The research group examined niches for promoting metastatic colonization using the generation of human-in-mouse ovarian cancer xenograft models in immunodeficient mice. Pathological examinations revealed the existence of S100A4-negative and CD90-positive stem-like cells in vimentin-positive normal neighboring alveolar epithelial cells. |
11 |
|
Salahudeen AA, Choi SS, Rustagi A, Zhu J, van Unen V, de la O SM, Flynn RA, Margalef-Català M, Santos AJM, Ju J, Batish A, Usui T, Zheng GXY, Edwards CE, Wagar LE, Luca V, Anchang B, Nagendran M, Nguyen K, Hart DJ, Terry JM, Belgrader P, Ziraldo SB, Mikkelsen TS, Harbury PB, Glenn JS, Garcia KC, Davis MM, Baric RS, Sabatti C, Amieva MR, Blish CA, Desai TJ, Kuo CJ. Progenitor identification and SARS-CoV-2 infection in human distal lung organoids. Nature 2020; 588: 670-675. |
Lung organoid scRNA-seq at time points before ciliated differentiation identified expression of the SARS-CoV-2 receptor ACE2 and processing protease TMPRSS2 mRNAs predominantly in progenitor or stem-like cells of alveolar type 2. Reseach group demonstrated functional heterogeneity among basal cells and establishes a facile in vitro organoid model of human distal lung infections, including COVID-19-associated pneumonia. |
12 |
|
Hayashi T, Sano K, Konishi I. Possibility of SARS-CoV-2 infection in metastatic microenvironment of cancer. Molecules at Play in Cancer: Curr. Issues Mol. Biol. 2022, 44(1), 233-241 |
In the pulmonary micrometastatic niche of patients with ovarian cancer, alveolar epithelial stem-like cells were found adjacent to the ovarian cancer. Moreover, angiotensin-converting enzyme 2, a host-side receptor for SARS-CoV-2, was expressed in these alveolar epithelial stem-like cells. Furthermore, the spike glycoprotein receptor-binding domain of SARS-CoV-2 bound to alveolar epithelial stem-like cells. The prevention of de novo niche formation in metastatic diseases might constitute a new strategy for the clinical treatment of COVID-19 for patients with cancer. |
13 |

This manuscript is a resubmission of an earlier submission. The following is a list of the peer review reports and author responses from that submission.
Round 1
Reviewer 1 Report
Manuscript #: biomedicines-1933388
Title: Pathological evidence for residual SARS-CoV-2 in the micrometastatic niche of a patient with ovarian cancer
Authors: Hayashi et al.
The above case report is strange. In the introduction, the authors stated that the severity and mortality of COVID-19 in patients with ovarian cancer is higher than in patients without cancer. Then they described the case of patient with ovarian cancer associated with “malignant left ovarian tumor torsion” and coexisting infection with SARSpCoV-2. As described by the authors, the patient underwent “laparoscopic left and right uterine adnexal tumor resection”. The reason why intraoperative histological diagnosis and hysterectomy with bilateral adnexectomy was not made is unknown. Next, in immunohistochemical examination of tissues, the authors found “expression of angiotensin-converting enzyme 2 (ACE2), a host-side receptor for SARS-CoV-2, in ovarian tissue cells in contact with the micrometastatic niche of high-grade serous ovarian cancer (HG-SOC), and SARS-CoV-2”. This observation led the authors to conclusion that the increased severity of COVID-19 in patients with ovarian cancer is a result of ACE2 presence in ovarian tissue “in contact with micrometastatic niche”.
It should be stated that the manuscript contains numerous and factual and conceptive errors, and the conclusion is strictly speculative.
The major errors include:
- The severity of COVID-19 depends on gender, age, comorbidities, and genetic factors. Previous studies also found that the severity of COVID-19 is increased in the presence of cancer (reference 1). Ovarian cancer is most likely to be associated with increased severity of COVID-19, but the article mentioned by the authors (reference 1) does not provide such an association. It deals with cancers as such without establishing an individual relationship between COVID-19 and ovarian cancer. For this reason, reference 1 does not supports the authors’ hypothesis.
- The authors should know that ovarian cancer is associated with immune deficiencies leading to progression of the tumor in the host. These effect are associated with the presence of regulatory T cells, the inhibition of natural killer cytotoxic responses, the accumulation of myeloid suppressor cells in the tumor, deficiencies on interferon signaling, the secretion of cytokines that enhance tumor growth (foe example, IL-6, IL-10, CSF-1, TGF-b, TNF), and the expression of surface molecules (for example, HLA-G, B7-H1, B7-H4, CD40, CD80) that have a role on immune suppression (PMID: 19910891). These observations led to steps to increase the immune system’s ability to fight ovarian cancer (PMID: 33123161; PMID: 25894333).
- The authors should know that renin-angiotensin-aldosterone system (RAAS) is a vital system of human body, as it maintains plasma sodium concentration, arterial blood pressure, extracellular volume, and adequate blood flow through the cerebral and coronary circulation. Previous studies have shown that ACE2 and TMPRSS2 are expressed in normal condition in the lung, gastrointestinal tract, heart, kidney, and ovary (PMID: 33953595; PMID: 344184; PMID: 32365180). These observations indicate that ACE2 is present in the ovary and lung without ovarian cancer and metastases to the lung. For this reason, the hypothesis of authors is worthless.
- The authors should also know that numerous studies indicate that viral components (RNA, proteins) of SARS-CoV-2 can be found in multiple organs such as the pharynx, trachea, lungs, heart, vessels, intestines, brain, male genitals and kidneys, as well as in body fluids such as mucus, blood, saliva, urine, cerebrospinal fluid, semen and breast milk (PMID: 33125439).
Author Response
Manuscript ID biomedicines-1933388
Reviewer 1:
Authors: Hayashi et al.
The above case report is strange. In the introduction, the authors stated that the severity and mortality of COVID-19 in patients with ovarian cancer is higher than in patients without cancer. Then they described the case of patient with ovarian cancer associated with “malignant left ovarian tumor torsion” and coexisting infection with SARSp-CoV-2. As described by the authors, the patient underwent “laparoscopic left and right uterine adnexal tumor resection.” The reason why intraoperative histological diagnosis and hysterectomy with bilateral adnexectomy was not made is unknown. Next, in immunohistochemical examination of tissues, the authors found “expression of angiotensin-converting enzyme 2 (ACE2), a host-side receptor for SARS-CoV-2, in ovarian tissue cells in contact with the micrometastatic niche of high-grade serous ovarian cancer (HG-SOC), and SARS-CoV-2”. This observation led the authors to conclusion that the increased severity of COVID-19 in patients with ovarian cancer is a result of ACE2 presence in ovarian tissue “in contact with micrometastatic niche”.
It should be stated that the manuscript contains numerous and factual and conceptive errors, and the conclusion is strictly speculative.
The major errors include:
Comment 1. The severity of COVID-19 depends on gender, age, comorbidities, and genetic factors. Previous studies also found that the severity of COVID-19 is increased in the presence of cancer (reference 1). Ovarian cancer is most likely to be associated with increased severity of COVID-19, but the article mentioned by the authors (reference 1) does not provide such an association. It deals with cancers as such without establishing an individual relationship between COVID-19 and ovarian cancer. For this reason, reference 1 does not supports the authors’ hypothesis.
Answer 1. We appreciate your comment. We particularly agree with the following comments from the reviewer.
Comment from reviewer 1. Previous studies also found that the severity of COVID-19 is increased in the presence of cancer (reference 1). Ovarian cancer is most likely to be associated with increased severity of COVID-19, but the article mentioned by the authors (reference 1) does not provide such an association.
We selected Reference 1 to help nonspecialist readers understand that the severity of COVID-19 increases in the presence of cancer. However, to the best of our knowledge, there are no clinical trial results indicating that ovarian cancer is associated with increased COVID-19 severity. Our medical staff mainly provides gynecological care. They believe that the pathological findings of patients with advanced and/or recurrent ovarian cancer infected with SARS-CoV-2 may help decrease COVID-19 severity in patients with other cancer types. We trust that the reviewers will understand why we chose Reference 1.
At the end of the Introduction section, we have stated the purpose of our research:
Our research aimed to identify the cause of increasing COVID-19 severity in cancer patients using molecular pathological studies. The primary endpoint of our clinical research was to show that the severity of COVID-19 in cancer patients can be attributed to these pathological features.
Comment 2. The authors should know that ovarian cancer is associated with immune deficiencies leading to progression of the tumor in the host. These effect are associated with the presence of regulatory T cells, the inhibition of natural killer cytotoxic responses, the accumulation of myeloid suppressor cells in the tumor, deficiencies on interferon signaling, the secretion of cytokines that enhance tumor growth (foe example, IL-6, IL-10, CSF-1, TGF-b, TNF), and the expression of surface molecules (for example, HLA-G, B7-H1, B7-H4, CD40, CD80) that have a role on immune suppression (PMID: 19910891). These observations led to steps to increase the immune system’s ability to fight ovarian cancer (PMID: 33123161; PMID: 25894333).
Answer 2. We appreciate your comment regarding the relationship between immune function alterations and cancer cell proliferation in patients with ovarian cancer. The comments from the reviewers include content that affects the life prognosis of ovarian cancer patients infected with SARS-CoV-2. Therefore, we have added the following comments from the reviewers to the Discussion section in the revised manuscript.
The following content has been added to the Discussion section
Recent research demonstrated that ovarian cancer is associated with immune deficiencies leading to tumor progression in the host. These effects are associated with the presence of regulatory T cells, inhibition of natural killer cytotoxic responses, accumulation of myeloid suppressor cells in the tumor, deficiencies on interferon signaling, secretion of cytokines that enhance tumor growth (e.g., IL-6, IL-10, CSF-1, TGF-b, TNF), and expression of surface molecules (e.g., HLA-G, B7-H1, B7-H4, CD40, CD80) that play a role in immune suppression (20). Presumably, the weakened immune function observed in ovarian cancer patients affects the growth of ovarian cancer (21,22) and, in turn, the infectivity of SARS-CoV-2 for ovarian cancer patients. However, our pathological studies demonstrated the increased expression of ACE2 and SARS-CoV-2 virus particles in the vicinity of ovarian tissue cells that are in contact with the micrometastatic niche. The cytokines secreted by tumor cells and/or ovarian tissue cells that are in contact with the micrometastatic niche are thought to induce upregulation of ACE2 expression. Further clinical studies are required to elucidate the association between cytokine expression observed in ovarian cancer patients and SARS-CoV-2 infectivity.
- Torres MP, Ponnusamy MP, Lakshmanan I, Batra Immunopathogenesis of ovarian cancer. Minerva Med. 2009 Oct;100(5):385-400.
- Yang C, Xia BR, Zhang ZC, Zhang YJ, Lou G, Jin WL. Immunotherapy for ovarian cancer: Adjuvant, combination, and neoadjuvant. Front Immunol. 2020 Oct 6;11:577869.
doi: 10.3389/fimmu.2020.577869.
- Santoiemma PP, Powell DJ Jr. Tumor infiltrating lymphocytes in ovarian cancer. Cancer Biol Ther. 2015;16(6):807-20. doi: 10.1080/15384047.2015.1040960.
We aimed to identify the cause of increasing COVID-19 severity in cancer patients. We found increased ACE2 expression and SARS-CoV-2 virus particles in the vicinity of ovarian tissue cells that are in contact with the micrometastatic niche. Thus, we wrote a manuscript about a patient with ovarian cancer found to have SARS-CoV-2 infection in the vicinity of a metastatic lesion as a case report.
Our medical staff provided emergency surgical treatment for the patient. After surgical treatment, we were unable to collect biological/medical samples, such as blood, for clinical studies to obtain immunological findings. We trust that the reviewers will understand that urgent surgical intervention was given to save the patient’s life.
Comment 3. The authors should know that renin-angiotensin-aldosterone system (RAAS) is a vital system of human body, as it maintains plasma sodium concentration, arterial blood pressure, extracellular volume, and adequate blood flow through the cerebral and coronary circulation. Previous studies have shown that ACE2 and TMPRSS2 are expressed in normal condition in the lung, gastrointestinal tract, heart, kidney, and ovary (PMID: 33953595; PMID: 344184; PMID: 32365180). These observations indicate that ACE2 is present in the ovary and lung without ovarian cancer and metastases to the lung. For this reason, the hypothesis of authors is worthless.
Answer 3. We sincerely thank the reviewers for their very valuable comments.
We carefully read the research reports. The PMID: 33953595: report shows the expression of ACE2 in human lung tissue. The PMID: 344184: report is rather old and its contents cannot be confirmed.
The PMID: 32365180: report shows that the expression of ACE2 mRNA in oocytes is relatively high; thus, the ovary and oocyte are considered potential targets of 2019-nCoV. The PMID: 32365180: report also shows that ACE2 is present in stroma and granulosa cells as well as oocytes in immature rat ovaries, the expression of which is enhanced in the antral and preovulatory follicles subjected to equine CG treatment. The PMID: 32365180: report describes the expression of ACE2 mRNA in rat reproductive organs. Furthermore, the PMID: 32365180: report does not present any images or photographs showing the expression of ACE2 in each tissue of human ovary. Furthermore, in the PubMed website, the comment “COVID-19 and human reproduction: hypothesis needs to be investigated” is presented in a formal manner to the PMID: 32365180: report.
The results of the clinical research conducted by our clinical research group indicated that the expressions of human ACE2 were not detected in the follicle and stromal cells as well as cortex in healthy human ovary tissues, as presented in S. Figure 5.
The international database, i.e., the human protein atlas, also demonstrated that the expressions of human ACE2 were not detected in follicle and stroma cells in healthy human ovary tissues.
https://www.proteinatlas.org/ENSG00000130234-ACE2/tissue/ovary
PMID: 33953595; Factors associated with the expression of ACE2 in Human Lung Tissue: Pathological Evidence from Patients with Normal FEV1 and FEV1/FVC. J Inflamm Res. 2021, 14:1677-1687.
PMID: 344184; Hospitals. 1978 Apr 16;52(8):65.
PMID: 32365180; Potential influence of COVID-19/ACE2 on the female reproductive system. Mol Hum Reprod. 2020 Jun 1;26(6):367-373.
Comment 4. The authors should also know that numerous studies indicate that viral components (RNA, proteins) of SARS-CoV-2 can be found in multiple organs such as the pharynx, trachea, lungs, heart, vessels, intestines, brain, male genitals and kidneys, as well as in body fluids such as mucus, blood, saliva, urine, cerebrospinal fluid, semen and breast milk (PMID: 33125439).
Answer 4. We sincerely thank the reviewers for their very valuable comments. We carefully read the research report (PMID: 33125439). We understand well that the viral components (RNA, proteins) of SARS-CoV-2 can be found in multiple organs, such as the pharynx, trachea, lungs, heart, vessels, intestines, brain, male genitals, and kidneys, as well as in body fluids, such as mucus, blood, saliva, urine, cerebrospinal fluid, semen, and breast milk. Therefore, to confirm the presence of SARS-CoV-2 in the ovarian tissue extracted from the patients, we examined the presence or absence of SARS-CoV-2 in the excised ovarian tissue via electron microscopy.
As presented in Figure 3 E and F, the photographs of the excised ovarian tissue taken via electron microscopy clearly show the presence of SARS-CoV-2 viral particles.

Reviewer 2 Report
Sample size is very small. Level of ILs and cytokines in the tissue may be provided to enrich the strength of the manuscript.
Hayashi et al. are attempting to identify the cause of COVID-19 severity in cancer patients. They discovered the increased expression of ACE2 and SARS-CoV-2 virus particles in the vicinity of ovarian tissue cells in contact with the micrometastatic niche. The piece of work is unique and well-written. However, the sample size is extremely little, and quantifying the levels of ILs and cytokines in the same may enhance the manuscript's strength.
Author Response
Manuscript ID biomedicines-1933388
Reviewer 2:
Comments and Suggestions for Authors
Comment 1. Sample size is very small. Level of ILs and cytokines in the tissue may be provided to enrich the strength of the manuscript.
Answer 1. We appreciate and agree with your comment.
We certainly think your comments are important. In patients with cancer, we found ACE2 expression in cells that are in contact with metastatic tumors detected in organs and tissues that did not express ACE2, the receptor for SARS-CoV-2. Therefore, for these patients, we confirmed SARS-CoV-2 infection in cells that are in contact with metastatic tumors. We wrote a manuscript about a patient with ovarian cancer found to have SARS-CoV-2 infection in the vicinity of a metastatic lesion as a case report.
We appreciate your comment regarding the relationship between immune function alterations and cancer cell proliferation in patients with ovarian cancer. The comments from the reviewers include content that affects the life prognosis of ovarian cancer patients infected with SARS-CoV-2. Therefore, we have added the following comments from the reviewers to the discussion section in the revised manuscript.
The following content has been added to the Discussion section
Recent research demonstrated that ovarian cancer is associated with immune deficiencies leading to tumor progression in the host. These effects are associated with the presence of regulatory T cells, inhibition of natural killer cytotoxic responses, accumulation of myeloid suppressor cells in the tumor, deficiencies on interferon signaling, secretion of cytokines that enhance tumor growth (e.g., IL-6, IL-10, CSF-1, TGF-b, TNF), and expression of surface molecules (e.g., HLA-G, B7-H1, B7-H4, CD40, CD80) that play a role in immune suppression (20). Presumably, the weakened immune function observed in ovarian cancer patients affects the growth of ovarian cancer (21,22) and, in turn, the infectivity of SARS-CoV-2 to ovarian cancer patients. On the other hand, our pathological studies demonstrated the increased expression of ACE2 and SARS-CoV-2 virus particles in the vicinity of ovarian tissue cells that are in contact with the micrometastatic niche. The cytokines secreted by the tumor cells and/or ovarian tissue cells that are in contact with the micrometastatic niche are thought to induce upregulation of ACE2 expression. Further clinical studies are required to elucidate the association between cytokine expression observed in patients with ovarian cancer and SARS-CoV-2 infectivity.
- Torres MP, Ponnusamy MP, Lakshmanan I, Batra Immunopathogenesis of ovarian cancer. Minerva Med. 2009 Oct;100(5):385-400.
- Yang C, Xia BR, Zhang ZC, Zhang YJ, Lou G, Jin WL. Immunotherapy for ovarian cancer: Adjuvant, combination, and neoadjuvant. Front Immunol. 2020 Oct 6;11:577869.
doi: 10.3389/fimmu.2020.577869.
- Santoiemma PP, Powell DJ Jr. Tumor infiltrating lymphocytes in ovarian cancer. Cancer Biol Ther. 2015;16(6):807-20. doi: 10.1080/15384047.2015.1040960.
Comment 2. Hayashi et al. are attempting to identify the cause of COVID-19 severity in cancer patients. They discovered the increased expression of ACE2 and SARS-CoV-2 virus particles in the vicinity of ovarian tissue cells in contact with the micrometastatic niche. The piece of work is unique and well-written. However, the sample size is extremely little, and quantifying the levels of ILs and cytokines in the same may enhance the manuscript's strength.
Answer 2. We appreciate and particularly agree with the following comments from the reviewer.
Comment from reviewer 2. However, the sample size is extremely little, and quantifying the levels of ILs and cytokines in the same may enhance the manuscript's strength.
Our medical staff provided emergency surgical treatment for a patient with ovarian cancer infected with SARS-CoV-9. After surgical treatment, we were unable to collect biological/medical samples, such as blood, for clinical studies to obtain immunological findings. We trust that the reviewers will understand that urgent surgical intervention was given to save the patient’s life.
The next time we encounter a similar case and our medical staff provide surgical treatment, we will take samples of body fluids, such as blood, from the patient for immunological analysis. Further clinical studies are required to elucidate the association between cytokine expression observed in patients with ovarian cancer and SARS-CoV-2 infectivity.

Reviewer 3 Report
In this case report, Hayashi and colleagues, examined the infection status of SARS-CoV-2 in primary tumor and micrometastasis tissues of a 47 years old patient suffering from ovarian cancer and COVID-19. Main results indicate that angiotensin-converting enzyme 2 (ACE2) expression was observed, and SARS-CoV-2 particles were detected in ovarian tissue cells in contact with the micrometastatic niche of high-grade serous ovarian cancer.
1) Given the fact that this is a case report study, i.e., only one patient is described, “patients” should be singular instead of plural in the abstract
2) This in an interesting work and well performed in terms of analyses conducted. Indeed, it underlines the importance in managing COVID-19 in ovarian cancer affected patients. However, th eintroduction is too concise and should be improved by including notions on the implication of SARS-CoV-2 and the development/progression of different cancer types, both solid tumors and hematological neoplasms, authors can check doi: 10.1016/j.esmoop.2022.100403, doi: 10.1007/s00520-021-06175-z and doi: 10.1007/s12254-021-00741-1. The rationale sohuld be well described, as well.
3) Section 3, “On May 15, 2022, the patient underwent a nasopharyngeal swab and was confirmed positive for SARS-CoV-2 via a reverse transcription-polymerase chain reaction (RT-PCR) test followed by treatment.” -->. This reference describing different SARS-CoV-2 detection methods should be included DOI: 10.3390/microorganisms10061193
4) References should be included in the methods
5) Methods, acronyms should be avoided in the sub head titles
6) introduction “Hence, we report that more severe COVID-19 cases in patients with cancer may be attributed to these pathological features.” This is not the aim but it seems more like a conclusion. At the end of the introduction the authors should clearly describe the purpose of the work.
7) A conclusive paragraph at the end of the discussion should be included.
8) Figure titles should be included. Moreover, the word “figure” at the top-left of the figures should be removed
9) Authors should carefully check the font style and the presence of background colors. E.g., “…By contrast, the expression of ACE2 and spike protein was…”
10) Table 1 should be included after the first time being mentioned
11) Acronyms should be carefully checked in order to avoid repetitions, for instance “Immunohistochemistry (IHC)”
Author Response
Manuscript ID biomedicines-1933388
Reviewer 3:
Comments and Suggestions for Authors
In this case report, Hayashi and colleagues, examined the infection status of SARS-CoV-2 in primary tumor and micrometastasis tissues of a 47 years old patient suffering from ovarian cancer and COVID-19. Main results indicate that angiotensin-converting enzyme 2 (ACE2) expression was observed, and SARS-CoV-2 particles were detected in ovarian tissue cells in contact with the micrometastatic niche of high-grade serous ovarian cancer.
Comment 1. Given the fact that this is a case report study, i.e., only one patient is described, “patients” should be singular instead of plural in the abstract.
Answer 1. We appreciate and agree with the comments from the reviewer. We have rewritten part of the Abstract section in the revised manuscript according to the reviewer’s instructions as follows:
Abstract: In previous clinical studies, severe acute respiratory syndrome coronavirus 2 (SARS-CoV-2) infection in cancer patients has a high risk of aggravation and mortality than in healthy infected individuals. Inoculation with COVID-19 vaccine reduces the risk of SARS-CoV-2 infection and COVID-19 severity. However, vaccination-induced anti-SARS-CoV-2 antibody production is said to be lower in cancer patients than in healthy individuals. In addition, the rationale for why the condition of patients with cancer worsens with COVID-19 is not well understood. Therefore, we examined the infection status of SARS-CoV-2 in the primary tumor and micrometastasis tissues of the patient patients with cancer and COVID-19. In this study, the expression of angiotensin-converting enzyme 2 (ACE2) was observed, and SARS-CoV-2 patient were detected SARS-CoV-2 particles was detected in ovarian tissue cells in contact with the micrometastatic niche of the patient with high-grade serous ovarian cancer. We believe that the severity of COVID-19 in patients with cancer can be attributed to these pathological features. Therefore, the pathological findings of patients with advanced and recurrent ovarian cancer infected with SARS-CoV-2 may help decrease COVID-19 severity in patients with other cancer types.
Comment 2. This in an interesting work and well performed in terms of analyses conducted. Indeed, it underlines the importance in managing COVID-19 in ovarian cancer affected patients. However, the introduction is too concise and should be improved by including notions on the implication of SARS-CoV-2 and the development/progression of different cancer types, both solid tumors and hematological neoplasms, authors can check doi: 10.1016/j.esmoop.2022.100403, doi: 10.1007/s00520-021-06175-z and doi: 10.1007/s12254-021-00741-1. The rationale should be well described, as well.
Answer 2. We appreciate and agree with the comments from the reviewer. We carefully checked doi: 10.1016/j.esmoop.2022.100403, doi: 10.1007/s00520-021-06175-z, and doi: 10.1007/s12254-021-00741-1. We have also rewritten part of the Abstract and Introduction sections in the revised manuscript according to the reviewer’s instructions as follows:
The following content has been added to the Introduction section
Patients with hematological malignancies or solid tumors who have been infected with SARS-CoV-2 are at an increased risk of thromboembolism and associated complications, such as pulmonary vascular occlusive thromboinflammatory syndrome (5). For hospitalized patients with cancer, prophylaxis using low-molecular-weight heparin (LMWH) is recommended for cancer patients infected with SARS-CoV-2 (5). COVID-19 vaccination in healthy individuals reduces COVID-19 severity. Host immune responses to the SARS-CoV-2 vaccine may be suboptimal in patients with solid tumors, whereas SARS-CoV-2 vaccination-induced antibody productions are further reduced in patients with hematological malignancies (6). The seroconversion rate of patients with hematological malignancies after the second dose of COVID-19 vaccine is approximately 80%, which is lower than the 98% seroconversion rate observed in solid tumor patients (6).
As a preventive measure against SARS-CoV-2 infection, hygiene measures, physical distancing, and wearing of face masks are recommended for patients with cancer, intimate family members, and caregivers (5). Furthermore, there may be a need to apply common preventive strategies, such as letting cancer patients stay in private rooms (a). To reduce clinic visits, routine SARS-CoV-2 swab testing, vaccination of healthcare workers working closely with cancer patients as well as caregivers, and efforts to reorganize the hematology unit with telemedicine are supported (5). Summarizing the results of clinical studies so far, patients with cancer, especially those with hematological malignancies, have a higher risk of developing COVID-19 and more severe outcomes. Hygiene precautions should be taken as early as possible during pandemic (7).
- Buske C, Dreyling M, Alvarez-Larrán A, Apperley J, Arcaini L, Besson C, et al. Managing hematological cancer patients during the COVID-19 pandemic: an ESMO-EHA interdisciplinary expert consensus. ESMO Open. 2022;7(2). doi: 10.1016/j.esmoop.2022.100403 PMID: 35272130
- Zojer N. SARS-CoV-2 vaccination in patients with solid tumors or hematological malignancies: Is the pandemic over for fully vaccinated patients? 2021;14:221-223. doi: 10.1007/s12254-021-00741-1 PMID: 3454888
- Goudsmit A, Cubilier E, Meert AP, Aftimos P, Stathopoulos K, Spilleboudt C, et al. Factors associated with SARS-CoV-2 infection and outcome in patients with solid tumors or hematological malignancies: A single-center study. Support Care Cancer. 2021;29:6271-6278. doi: 10.1007/s00520-021-06175-z PMID: 33851236
Comment 3. Section 3, “On May 15, 2022, the patient underwent a nasopharyngeal swab and was confirmed positive for SARS-CoV-2 via a reverse transcription-polymerase chain reaction (RT-PCR) test followed by treatment.” -->. This reference describing different SARS-CoV-2 detection methods should be included DOI: 10.3390/microorganisms10061193
Answer 3. We appreciate and agree with the comments from the reviewer. We have added the reference “DOI: 10.3390/microorganisms10061193” in the revised manuscript according to the reviewer’s instructions.
On May 15, 2022, the patient underwent a nasopharyngeal swab and was confirmed positive for SARS-CoV-2 via a reverse transcription-polymerase chain reaction (RT-PCR) test followed by treatment (10).
- Rotondo JC, Martini F, Maritati M, Caselli E, Gallenga CE, Guarino M, et al. Advanced molecular and immunological diagnostic methods to detect SARS-CoV-2 infection. Microorganisms. 2022;10. doi: 10.3390/microorganisms10061193 PMID: 35744711
Comment 4. References should be included in the methods
Answer 4. We appreciate and agree with the comments from the reviewer. We have added the references in the Methods section in the revised manuscript according to the reviewer’s instructions.
2.2. Immunohistochemistry (IHC).
IHC was performed using normal methods with the primary and second antibodies conjugated with immunofluorescence as previously described (8).
2.4. Statistical Analysis.
For comparing two groups, unpaired two-tailed t-test or Mann–Whitney U test was used. Multiple comparisons were conducted via one-way analysis of ANOVA with Tukey’s post hoc test or Kruskal–Wallis analysis with post hoc Steel–Dwass or Steel test as previously described (9).
- Hayashi T, Sano K, Konishi I. Possibility of SARS-CoV-2 infection in the metastatic microenvironment of cancer. Curr Issues Mol Biol. 2022;44(1):233-241. doi: 3390/cimb44010017
- Hayashi T, Ichikawa M, Konishi I. Spontaneous myocarditis in mice predisposed to autoimmune disease: Including vaccination-induced onset. 2022 Jun 18;10(6):1443. doi: 10.3390/biomedicines10061443
Comment 5. Methods, acronyms should be avoided in the sub head titles
Answer 5. We appreciate and agree with the comments from the reviewer. We have deleted the acronyms in the Methods section in the revised manuscript according to the reviewer’s instructions.
Comment 6. introduction “Hence, we report that more severe COVID-19 cases in patients with cancer may be attributed to these pathological features.” This is not the aim but it seems more like a conclusion. At the end of the introduction the authors should clearly describe the purpose of the work.
Answer 6. We appreciate and agree with the comments from the reviewer. At the end of the Introduction section, we described the purpose of the work in the revised manuscript according to the reviewer’s instructions as follows:
At the end of the Introduction section, we have stated the purpose of our research as follows:
The pathological features of COVID-19, especially in patients with recurrent cancer or cancer patients with relapse or metastasis, remain largely unknown. In our recent study, the expression of angiotensin-converting enzyme 2 (ACE2), a host-side receptor for SARS-CoV-2, was observed in ovarian tissue cells that are in contact with the micrometastatic niche of high-grade serous ovarian cancer (HG-SOC). In addition, the possibility of SARS-CoV-2 infection was observed in ovarian tissue cells in which such an ACE2 expression was detected (8). Our research aimed to identify the cause of increasing COVID-19 severity in cancer patients using molecular pathological studies. The primary endpoint of our clinical research is to show that the severity of COVID-19 in cancer patients can be attributed to these pathological features. Hence, we report that more severe COVID-19 cases in patients with cancer may be attributed to these pathological features.
Comment 7. A conclusive paragraph at the end of the discussion should be included.
Answer 7. We appreciate and agree with the comments from the reviewer. We included a conclusive paragraph at the end of the Discussion section of the revised manuscript according to the reviewer’s instructions as follows:
The following content has been added to the Discussion section
Recent research demonstrated that ovarian cancer is associated with immune deficiencies leading to tumor progression in the host. These effects are associated with the presence of regulatory T cells, inhibition of natural killer cytotoxic responses, accumulation of myeloid suppressor cells in the tumor, deficiencies on interferon signaling, secretion of cytokines that enhance tumor growth (e.g., IL-6, IL-10, CSF-1, TGF-b, TNF), and expression of surface molecules (e.g., HLA-G, B7-H1, B7-H4, CD40, CD80) that play a role in immune suppression (20). Presumably, the weakened immune function observed in ovarian cancer patients affects the growth of ovarian cancer (21,22) and, in turn, the infectivity of SARS-CoV-2 to ovarian cancer patients. On the other hand, our pathological studies demonstrated the increased expression of ACE2 and SARS-CoV-2 virus particles in the vicinity of ovarian tissue cells that are in contact with the micrometastatic niche. The cytokines secreted by ovarian tissue cells that are in contact with the micrometastatic niche are thought to induce upregulation of ACE2 expression. Further clinical studies are required to elucidate the association between cytokine expression observed in patients with ovarian cancer and SARS-CoV-2 infectivity.
- Torres MP, Ponnusamy MP, Lakshmanan I, Batra Immunopathogenesis of ovarian cancer. Minerva Med. 2009 Oct;100(5):385-400.
- Yang C, Xia BR, Zhang ZC, Zhang YJ, Lou G, Jin WL. Immunotherapy for ovarian cancer: Adjuvant, combination, and neoadjuvant. Front Immunol. 2020 Oct 6;11:577869.
doi: 10.3389/fimmu.2020.577869.
- Santoiemma PP, Powell DJ Jr. Tumor infiltrating lymphocytes in ovarian cancer. Cancer Biol Ther. 2015;16(6):807-20. doi: 10.1080/15384047.2015.1040960.
At the end of the Discussion section, we have stated the conclusive content as follows:
Our research demonstrated the increased expression of ACE2 and SARS-CoV-2 virus particles in the vicinity of ovarian tissue cells that are in contact with the micrometastatic niche. The pathological findings of patients with advanced and/or recurrent ovarian cancer infected with SARS-CoV-2 may help decrease COVID-19 severity in patients with other cancer types. In clinical studies, molecular pathological analysis in COVID-19-infected patients with other cancer types is needed. Moreover, a timely health examination follow-up for patients with cancer is strongly recommended in clinical practice.
Comment 8. Figure titles should be included. Moreover, the word “figure” at the top-left of the figures should be removed
Answer 8. We appreciate and agree with the comments from the reviewer. We included figure titles and removed the word “figure” at the top left of the figures in the revised manuscript according to the reviewer’s instructions.
Comment 9. Authors should carefully check the font style and the presence of background colors. E.g., “…By contrast, the expression of ACE2 and spike protein was…”
Answer 9. We appreciate and agree with the comments from the reviewer. We carefully checked the font style and the presence of background colors.
Comment 10. Table 1 should be included after the first time being mentioned
Answer 10. We appreciate and agree with the comments from the reviewer.
We included Table 1 after the first time of being mentioned in the revised manuscript according to the reviewer’s instructions.
We have placed Table 1 at the end of the Results section.
Comment 11. Acronyms should be carefully checked in order to avoid repetitions, for instance “Immunohistochemistry (IHC)”
Answer 11. We appreciate and agree with the comments from the reviewer. We carefully checked the acronyms to avoid repetitions.

Round 2
Reviewer 1 Report
Manuscript #: biomedicines-1933388 the second review
Title: Pathological evidence for residual SARS-CoV-2 in the micrometastatic niche of a patient with ovarian cancer
Authors: Hayashi et al.
In the new version of the manuscript, the authors have not been able to show what is the new in their manuscript. The case report concerns the patient with ovarian cancer coexisting infection with SARSpCoV-2. After surgery involving left and right uterine adnexal tumor resection (the description is unclear and it is unknown whether the operation also included hysterectomy or only a bilateral adnexectomy). In histological examination, the authors found “a high-grade serous carcinoma in the papillary nodule on the right outer surface of the left ovary. Serous tubal intraepithelial carcinoma (STIC) and high-grade serous carcinoma were are found in the left fimbriated end”. Immunohistochemistry have shown „a strong expression of ACE2 and cluster of differentiation 90 (CD90), in the ovary in some cells of the connective tissue/stroma cells in contact with or in the vicinity of the micrometastatic niche of HG-SOC On the other hand, the authors did not observed ACE2 expression in the connective tissue cells (stroma cells) other than the micrometastatic niche. In addition, the authors have found that “SARS-CoV-2 may be present in the HG-SOC tissue (?)”.Three days after the standard surgical treatment for ovarian tumors (including ovarian tumor torsion), the patient was stable and was transferred to a nearby general medical facility (supplementary files). The further fate of the patients is unknown. No data on the later course of COVID-19 or treatment of ovarian cancer.
The number of people infected with SARSpCoV-2 most likely exceeds a billion, and it is not surprise that also occurs in patients with ovarian cancer. In the previous review, the reviewer presented that it is already known that:
- ovarian cancer is associated with immune deficiencies leading to progression of the tumor in the host.
- renin-angiotensin-aldosterone system (RAAS) is a vital system of human body, and previous studies have shown that ACE2 and TMPRSS2 are expressed in normal condition in the lung, gastrointestinal tract, heart, kidney, and ovary These observations indicate that ACE2 is present in the ovary and lung without ovarian cancer and metastases to the lung.
- viral components (RNA, proteins) of SARS-CoV-2 can be found in multiple organs such as the pharynx, trachea, lungs, heart, vessels, intestines, brain, male genitals and kidneys, as well as in body fluids such as mucus, blood, saliva, urine, cerebrospinal fluid, semen and breast milk.
In addition:
- ACE expression was found in different clinical cancers, including ovarian cancer (PMID: 33821252).
- ACE2 and TMPRSS2are expressed in normal human ovaries (PMID: 34418496).
The references included in the previous and current above reviews were not intended to be included in the manuscript by the authors, but to make the authors aware of the lack of new observations.
Some other points:
In addition, it should be stated that, apart the lack of new data, the manuscript is chaotic and unclear. It is unknown what the last sentence in the abstract mean “Therefore, the pathological findings of patients with advanced and recurrent ovarian cancer infected with SARS-CoV-2 may help decrease COVID-19 severity in patients with other cancer types.”
Also it is not known what is a point of presenting Table 2 with minimal data of other patient, if the manuscript is a description of the case of one patient.
Case report, pathological findings, page 5. Patient’s histological data presented together with data from literature. This leads to chaos.
Author Response
Reviewer 1:
Comments and Suggestions for Authors
The number of people infected with SARSpCoV-2 most likely exceeds a billion, and it is not surprise that also occurs in patients with ovarian cancer. In the previous review, the reviewer presented that it is already known that:
- ovarian cancer is associated with immune deficiencies leading to progression of the tumor in the host.
- renin-angiotensin-aldosterone system (RAAS) is a vital system of human body, and previous studies have shown that ACE2 and TMPRSS2 are expressed in normal condition in the lung, gastrointestinal tract, heart, kidney, and ovary These observations indicate that ACE2 is present in the ovary and lung without ovarian cancer and metastases to the lung.
- viral components (RNA, proteins) of SARS-CoV-2 can be found in multiple organs such as the pharynx, trachea, lungs, heart, vessels, intestines, brain, male genitals and kidneys, as well as in body fluids such as mucus, blood, saliva, urine, cerebrospinal fluid, semen and breast milk.
In addition:
- ACE expression was found in different clinical cancers, including ovarian cancer (PMID: 33821252).
- ACE2 and TMPRSS2are expressed in normal human ovaries (PMID: 34418496).
The references included in the previous and current above reviews were not intended to be included in the manuscript by the authors, but to make the authors aware of the lack of new observations.
Some other points:
In addition, it should be stated that, apart the lack of new data, the manuscript is chaotic and unclear. It is unknown what the last sentence in the abstract mean “Therefore, the pathological findings of patients with advanced and recurrent ovarian cancer infected with SARS-CoV-2 may help decrease COVID-19 severity in patients with other cancer types.”
Also it is not known what is a point of presenting Table 2 with minimal data of other patient, if the manuscript is a description of the case of one patient.
Case report, pathological findings, page 5. Patient’s histological data presented together with data from literature. This leads to chaos.
- Comment from Reviewer 1.
In addition, it should be stated that, apart the lack of new data, the manuscript is chaotic and unclear. It is unknown what the last sentence in the abstract mean “Therefore, the pathological findings of patients with advanced and recurrent ovarian cancer infected with SARS-CoV-2 may help decrease COVID-19 severity in patients with other cancer types.”
- Answer:
We appreciate your comment. Our medical staff would like to make use of the importance of your comments in our future research.
Hayashi’s clinical research group is attempting to identify the cause of COVID-19 severity in cancer patients. We discovered the increased expression of ACE2 and SARS-CoV-2 viral particles in the vicinity of ovarian tissue cells in contact with the micrometastatic niche.
In some cancer types, in the micrometastatic niche, epithelial cells in contact with cancer cells reportedly differentiate into stem-like cells and progenitors. For example, the previous finding demonstrated that when prostate cancer cells metastasize to bone, they settle near stem cells in bone marrow, which promotes the development of a metastatic environment that supports tumor growth [1]. Our clinical research demonstrated that thus, stem cells and progenitors in contact with cancer cells in the micrometastatic niche are strongly induced to the express of ACE2 by cytokines released from cancer cells. The results obtained from our research show that SARS-CoV-2 infects stem cells and progenitors in contact with cancer cells in the micrometastatic niche.
- Shiozawa Y, Pedersen EA, Havens AM, Jung Y, Mishra A, Joseph J, Kim JK, Patel LR, Ying C, Ziegler AM, Pienta MJ, Song J, Wang J, Loberg RD, Krebsbach PH, Pienta KJ, Taichman RS. Human prostate cancer metastases target the hematopoietic stem cell niche to establish footholds in mouse bone marrow. J Clin Invest 2011; 121: 1298–1312.
We had our manuscripts reviewed by several Gynecology specialist doctor and Infectious disease specialist doctors before submission of our manuscript to medical journal.
We have received comments from experts, i.e., the members of the Advisory Committee of the World Obstetrics and the members of International Federation of Gynecology and Obstetrics (FIGO) Gynecology Association Oncology Committee, that our manuscript contains very new findings and makes good points to prevent the severity of COVID-19 in cancer patients.
We have also received comments from members of the Female Genital tumors classification on World Health Organization (WHO) that piece of work is unique and well-written.
Dr. Ikuo Konishi, who is coauthor, is a specialist in obstetrics and gynecology and a former director of the Japanese Society of Obstetrics and Gynecology and Dr. Konishi is also the director of Asian Gynecology Association, and Director of the Japanese Association of Medical Sciences.
Dr. Ikuo Konishi is the Advisory Committee of the World Obstetrics and International Federation of Gynecology and Obstetrics (FIGO) Gynecology Association Oncology Committee. Dr. Ikuo Konishi also is International Member of the Female Genital tumors classification on World Health Organization (WHO).
- Comment from Reviewer 1.
- viral components (RNA, proteins) of SARS-CoV-2 can be found in multiple organs such as the pharynx, trachea, lungs, heart, vessels, intestines, brain, male genitals and kidneys, as well as in body fluids such as mucus, blood, saliva, urine, cerebrospinal fluid, semen and breast milk.
- ACE2 and TMPRSS2are expressed in normal human ovaries (PMID: 34418496).
The references included in the previous and current above reviews were not intended to be included in the manuscript by the authors, but to make the authors aware of the lack of new observations.
- Answer:
We appreciate your comment. Our medical staff would like to make use of the importance of your comments in our future research.
Important Points from authors are indicated as followings.
The results of the clinical research conducted by our clinical research group indicated that the expressions of human ACE2 were not detected in the follicle and stromal cells as well as cortex in healthy human ovary tissues, as presented in S. Figure 5.
The international database, i.e., the human protein atlas, also demonstrated that the expressions of human ACE2 were not detected in follicle and stroma cells in healthy human ovary tissues.
https://www.proteinatlas.org/ENSG00000130234-ACE2/tissue/ovary
The Human Protein Atlas is a Swedish-based program initiated in 2003 with the aim to map all the human proteins in cells, tissues, and organs using an integration of various omics technologies, including antibody-based imaging, mass spectrometry-based proteomics, transcriptomics, and systems biology. All the data in the knowledge resource is open access to allow scientists both in academia and industry to freely access the data for exploration of the human proteome. The Human Protein Atlas consists of ten separate sections, each focusing on a particular aspect of the genome-wide analysis of the human proteins: https://www.proteinatlas.org/about
We sincerely thank the reviewers for their very valuable comments. We carefully read the research report (PMID: 33125439). We understand well that the viral components (RNA, proteins) of SARS-CoV-2 can be found in multiple organs, such as the pharynx, trachea, lungs, heart, vessels, intestines, brain, male genitals, and kidneys, as well as in body fluids, such as mucus, blood, saliva, urine, cerebrospinal fluid, semen, and breast milk. Therefore, to confirm the presence of SARS-CoV-2 in the ovarian tissue extracted from the patients, we examined the presence or absence of SARS-CoV-2 in the excised ovarian tissue via electron microscopy.
As presented in Figure 3 E and F, the photographs of the excised ovarian tissue taken via electron microscopy clearly show the presence of SARS-CoV-2 viral particles.
- Comment from Reviewer 1
ovarian cancer is associated with immune deficiencies leading to progression of the tumor in the host.
- Answer
We appreciate your comment regarding the relationship between immune function alterations and cancer cell proliferation in patients with ovarian cancer. The comments from the reviewers include content that affects the life prognosis of ovarian cancer patients infected with SARS-CoV-2. Therefore, we have added the following comments from the reviewers to the discussion section in the revised manuscript.
The following content has been added to the Discussion section
Recent research demonstrated that ovarian cancer is associated with immune deficiencies leading to tumor progression in the host. These effects are associated with the presence of regulatory T cells, inhibition of natural killer cytotoxic responses, accumulation of myeloid suppressor cells in the tumor, deficiencies on interferon signaling, secretion of cytokines that enhance tumor growth (e.g., IL-6, IL-10, CSF-1, TGF-b, TNF), and expression of surface molecules (e.g., HLA-G, B7-H1, B7-H4, CD40, CD80) that play a role in immune suppression (20). Presumably, the weakened immune function observed in ovarian cancer patients affects the growth of ovarian cancer (21,22) and, in turn, the infectivity of SARS-CoV-2 to ovarian cancer patients. On the other hand, our pathological studies demonstrated the increased expression of ACE2 and SARS-CoV-2 virus particles in the vicinity of ovarian tissue cells that are in contact with the micrometastatic niche. The cytokines secreted by the tumor cells and/or ovarian tissue cells that are in contact with the micrometastatic niche are thought to induce upregulation of ACE2 expression. Further clinical studies are required to elucidate the association between cytokine expression observed in patients with ovarian cancer and SARS-CoV-2 infectivity.
- Torres MP, Ponnusamy MP, Lakshmanan I, Batra Immunopathogenesis of ovarian cancer. Minerva Med. 2009 Oct;100(5):385-400.
- Yang C, Xia BR, Zhang ZC, Zhang YJ, Lou G, Jin WL. Immunotherapy for ovarian cancer: Adjuvant, combination, and neoadjuvant. Front Immunol. 2020 Oct 6;11:577869.
doi: 10.3389/fimmu.2020.577869.
- Santoiemma PP, Powell DJ Jr. Tumor infiltrating lymphocytes in ovarian cancer. Cancer Biol Ther. 2015;16(6):807-20. doi: 10.1080/15384047.2015.1040960.
- Comment from Reviewer 1.
- ACE expression was found in different clinical cancers, including ovarian cancer (PMID: 33821252).
- Answer
We appreciate your comment. Our medical staff would like to make use of the importance of your comments in our future research. Reviewer 1’s comment is consistent with new findings from our clinical studies.
In some cancer types, in the micrometastatic niche, epithelial cells in contact with cancer cells reportedly differentiate into stem-like cells and progenitors. For example, the previous finding demonstrated that when prostate cancer cells metastasize to bone, they settle near stem cells in bone marrow, which promotes the development of a metastatic environment that supports tumor growth [1]. Our clinical research demonstrated that thus, stem cells and progenitors in contact with cancer cells in the micrometastatic niche are strongly induced to the express of ACE2 by cytokines released from cancer cells. The results obtained from our research show that SARS-CoV-2 infects stem cells and progenitors in contact with cancer cells in the micrometastatic niche.
- Shiozawa Y, Pedersen EA, Havens AM, Jung Y, Mishra A, Joseph J, Kim JK, Patel LR, Ying C, Ziegler AM, Pienta MJ, Song J, Wang J, Loberg RD, Krebsbach PH, Pienta KJ, Taichman RS. Human prostate cancer metastases target the hematopoietic stem cell niche to establish footholds in mouse bone marrow. J Clin Invest 2011; 121: 1298–1312.
In the case of high grade serous ovarian cancer (HG-SOC), the primary cancer is STIC or carcinoma cells that have become malignant from epithelial cells of the fallopian tubes. Fallopian tube carcinoma cells and STIC metastasize to the ovary and proliferate well. The pathogenesis of such HG-SOC is shown in the S. Figure 8.
- Figure 8. SARS-CoV-2 infection in ovarian histiocytes in contact with the micrometastatic niche of HG-SOC. The origin of HG-SOC is STIC and/or epithelial malignancies in the fallopian tubes. STIC is most likely the precursor lesion of high-grade serous pelvic carcinomas, carcinosarcoma, and undifferentiated carcinoma. There is a window of 10 to 20 years between the development of abnormal cells, or lesions, in the fallopian tubes and the initiation of ovarian cancer. In the process of forming the micrometastatic niche in multiple organs, epithelial cells and stromal cells in contact with CTCs are initialized by secretory factors from CTCs that comprise the micrometastatic niche. ACE2 was possibly expressed in the epithelial stem-like cells or progenitor cells. ACE2 expression is observed, and SARS-CoV-2 particles are detected in ovarian histiocytes in contact with the micrometastatic niche of HG-SOC.

Reviewer 3 Report
The manuscript can be accepted in the present form. i suggest improving the quality of figure 2
Author Response
Manuscript ID biomedicines-1933388
Reviewer 3:
Comments and Suggestions for Authors
The manuscript can be accepted in the present form. I suggest improving the quality of figure 2.
Answer:
We appreciate your comment. We agree with your comment. According to your comment, we created and replaced Figure 2 in High quality as below.

Round 3
Reviewer 1 Report
Manuscript #: biomedicines-1933388 the third review.
Title: Pathological evidence for residual SARS-CoV-2 in the micrometastatic niche of a patient with ovarian cancer
Authors: Hayashi et al.
The case report is a kind of article, the aim of which is to present a new, unknown disease, or atypical course of an already known disease, as well as new unknown side effects in the course of the disease, or possible previously unknown side effects of the therapy used. The manuscript by Hayashi et al. does not meet either of these conditions. As shown in previous reviews, the data presented by the authors do not provide new, previously unknown observations, and no change in the shape of the manuscript will alter this fact. For this reason, the manuscript in the form of case report should be rejected. On the other hand, during the preparation of subsequent versions of the manuscript, the authors gained more and more knowledge about the impact of ovarian cancer, or more broadly cancer disease, on the efficiency of immune system, as well as the course of infectious disease, including COVID-19. Therefore, it seems that the authors could make an effective attempt to write a minireview article on this topic.